# Effect of Mn_x_O_y_ Nanoparticles Stabilized with Methionine on Germination of Barley Seeds (*Hordeum vulgare* L.)

**DOI:** 10.3390/nano13091577

**Published:** 2023-05-08

**Authors:** Andrey Blinov, Alexey Gvozdenko, Alexey Golik, Shahida A. Siddiqui, Fahrettin Göğüş, Anastasiya Blinova, David Maglakelidze, Irina Shevchenko, Maksim Rebezov, Andrey Nagdalian

**Affiliations:** 1Department of Physics and Technology of Nanostructures and Materials, Physical and Technical Faculty, North Caucasus Federal University, 355017 Stavropol, Russiaimshevchenko@ncfu.ru (I.S.); 2Department of Biotechnology and Sustainability, Technical University of Munich (TUM), 94315 Straubing, Germany; 3German Institute of Food Technologies (DIL e.V.), 49610 D-Quakenbrück, Germany; 4Department of Food Engineering, Engineering Faculty, University of Gaziantep, 27310 Gaziantep, Turkey; 5Biophotonics Center, Prokhorov General Physics Institute of the Russian Academy of Science, 119991 Moscow, Russia; 6Department of Scientific Research, V. M. Gorbatov Federal Research Center for Food Systems, 109240 Moscow, Russia; 7Laboratory of Food and Industrial Biotechnology, Faculty of Food Engineering and Biotechnology, North Caucasus Federal University, 355017 Stavropol, Russia

**Keywords:** nanoparticles, manganese dioxide, *Hordeum vulgare* L., seeds, germination energy, morphofunctional characteristics

## Abstract

The aim of this research was to study the effect of Mn_x_O_y_ nanoparticles stabilized with L-methionine on the morphofunctional characteristics of the barley (*Hordeum vulgare* L.) crop. Mn_x_O_y_ nanoparticles stabilized with L-methionine were synthesized using potassium permanganate and L-methionine. We established that Mn_x_O_y_ nanoparticles have a diameter of 15 to 30 nm. According to quantum chemical modeling and IR spectroscopy, it is shown that the interaction of Mn_x_O_y_ nanoparticles with L-methionine occurs through the amino group. It is found that Mn_x_O_y_ nanoparticles stabilized with L-methionine have positive effects on the roots and seedling length, as well as the seed germination energy. The effect of Mn_x_O_y_ nanoparticles on *Hordeum vulgare* L. seeds is nonlinear. At a concentration of 0.05 mg/mL, there was a statistically significant increase in the length of seedlings by 68% compared to the control group. We found that the root lengths of samples treated with Mn_x_O_y_ nanoparticle sols with a concentration of 0.05 mg/mL were 62.8%, 32.7%, and 158.9% higher compared to samples treated with L-methionine, KMnO_4_, and the control sample, respectively. We have shown that at a concentration of 0.05 mg/mL, the germination energy of seeds increases by 50.0% compared to the control sample, by 10.0% compared to the samples treated with L-methionine, and by 13.8% compared to the samples treated with KMnO_4_.

## 1. Introduction

Manganese oxide nanoparticles attract the interest of researchers due to unique physical, chemical, biological, and photocatalytic properties [1,2,3,4,5,6,7,8,9,10], ensuring its widespread use for the production of components applied in supercapacitors [11,12,13] as catalysts for the degradation of organic dyes [14,15,16] and for removal of heavy metal ions from water [17,18]. Manganese oxide nanoparticles and composites are widely used in medicine for the detection, diagnosis and treatment of cancer cells [19,20,21,22,23,24,25,26,27], and targeted chemotherapy [28]. Nanoparticles of manganese and its oxides (Mn_x_O_y_) can also be used in fertilizer components [29] and as bactericide preparation [30].

Another important area of application of nanomaterials, in particular, manganese nanoparticles and their oxides, is agriculture [31,32,33]. A large number of research papers are devoted to the study of the influence of nanoscale oxide forms on the morphofunctional characteristics of agricultural plants [34,35,36,37,38,39,40,41,42,43,44,45,46,47,48]. For example, the work of Margenot et al. [49] shows that the addition of copper oxide (CuO) nanoparticles increases the diameter of the roots of lettuce seedlings by 52% and the diameter of carrot seedlings by 26%. However, another form of this microelement did not show any significant effect. The effect of zinc oxide (ZnO) nanoparticles on the growth and antioxidant activity of chickpea seeding of the *Cicer arietinum* L. var. HC-1 culture was studied by Burman et al. [50]. It is shown that the addition of ZnO nanoparticles has a positive effect on the dry weight of shoots in comparison to the control sample microparticles of the inorganic form of zinc (ZnSO_4_), at a concentration of 1.5 ppm.

According to the FAO, there are about 1750 gene banks (seed banks) in the world, containing a total of 7.4 million agricultural seed samples [51]. These seeds have been stored for more than 10 years. At the same time, a number of studies have shown that these seeds have reduced germination energy and other morphofunctional characteristics during storage. For example, Maryam et al. proved that the germination percentage, germination energy, dry weight of seedlings and seed weight decrease significantly as seeds age [52]. Walters et al. studied wild and cultivated species from the collections of the gene bank of the US Department of Agriculture [53]. The study included 276 different species (codes W1–W276) with a total of 41,286 specimens collected from 1963 to 1968. The seeds were stored for 30–60 years, until 1977, at a temperature of +5 °C, and after that at −18 °C. The initial germination percentage, the actual germination percentage, and the number of years of storage were recorded and the half-life was calculated. The authors found that the germination percentage of old seeds averages from 40% to 60%.

Nanoparticles can be used to solve this problem. Younes et al. described the effect of silver nanoparticles (Ag NPs) on the germination of seven-year-old *Vicia faba* seeds [54]. The authors investigated the genotoxic effects associated with seed aging and were able to achieve an increase in root length and germination percentage of *Vicia faba* seeds. It is shown that when seeds are treated with 10 pm Ag NPs, there is an increase in the viability of seedlings of *Vicia faba* seeds. Yang et al. showed that titanium dioxide nanoparticles significantly increase the viability of old seeds and the formation of chlorophyll, stimulate the activity of ribulose 1,5-bisphosphate carboxylase, and enhance photosynthesis, thereby accelerating the growth and development of plants [55].

It is known that manganese-containing nanoparticles can be used in agriculture as fertilizers and for plant stress management [38,56,57,58,59,60]. Pradhan et al. [59] noted that manganese nanoparticles are the best source of the trace element Mn and have less toxicity to the *Vigna radiata* culture compared to the inorganic form (MnSO_4_). The use of Mn nanoparticles leads to an increase in the length of roots and sprouts by 40–70% compared to seeds treated with MnSO_4_ solution. Ye et al. [61] evaluated the effect of manganese nanoparticles on the germination of *Capsicum annuum* L. seeds in conditions of salt stress. It has been established that under salt stress, manganese nanoparticles behave as an important factor in the control of oxidative stress, contributing to the redistribution of trace elements between roots and shoots. Kasote et al. found that manganese oxide nanoparticles have lower phytotoxicity compared to other manganese-containing forms (KMnO_4_ and Mn_2_O_3_), change the chlorophyll and antioxidant profiles of seedlings, and had a noticeable effect on phenolic acids and phytohormones of watermelon seedlings at a concentration of 20 mg/L [62].

The purpose of this work was the synthesis and characterization of Mn_x_O_y_ nanoparticles stabilized with L-methionine, as well as the study of their effect on the morphofunctional characteristics of agricultural barley (*Hordeum vulgare* L.) seeds stored for more than 10 years. The choice of seeds with reduced morphofunctional characteristics is also due to the presence of a large amount of literature that uses standard seeds with normative and permissible morphofunctional characteristics to study the phytotoxicity of manganese compounds. However, at low manganese concentrations, as shown by Kasote et al. [62], Varaprasad et al. [63], and Wali et al. [64], toxic properties are not exhibited and metabolic processes are not normalized, being the main part of many manganese-containing enzymes, thus improving the morphofunctional parameters of seeds and young plants.

## 2. Materials and Methods

Reagent grade chemicals and grade A glassware were used in the present study. Conductivity of distilled water used was less than 1 µS/cm. All experiments were carried out in three-fold repetitions.

### 2.1. Synthesis of Mn_x_O_y_ Nanoparticles

The synthesis of nanoscale Mn_x_O_y_ stabilized with methionine was carried out according to the method presented in previous work [65]. Potassium permanganate, methionine, and distilled water were used for the synthesis. First, 0.005 g of L-methionine was weighed on analytical scales and put into 200 mL flask with distilled water. The resulting solution was stirred for a minute on an electromagnetic stirrer. In the second stage, the same process was completed with 0.005 g of potassium permanganate. At the third stage, two prepared solutions were mixed and stirred for 15 min. To study the effect of Mn_x_O_y_ nanoparticles on the seeds, only freshly prepared solutions were used. For research using diffractometry and scanning electron microscopy, the Mn_x_O_y_ samples were centrifuged at 15,000 rpm for 10 min and dried at 50 °C (2 h).

### 2.2. Samples Measurement

Hydrodynamic radius of synthesized Mn_x_O_y_ stabilized with L-methionine was measured with photon correlation spectrometer Photocor Complex (Photocor, Moscow, Russia).

The morphology of Mn_x_O_y_ particles was studied using a Mira-LMH scanning electron microscope (Tescan, Brno, Czech Republic). On a standard instrument table (12 mm), a double-sided conductive carbon tape was set. A thoroughly mixed Mn_x_O_y_ sample was taken from a dry powder ground in a mortar and applied to the tape. Then, carbon deposition of the order of 10 nm was performed on the QR 150 sputtering system and gas (nitrogen) was introduced into the microscope system by opening the balloon.

Parameters of the measurement:Accelerating voltage 10 kV;Working distance (WD) 4.9 mm;In-beam secondary electron detector.

X-ray powder diffraction (XRD) analysis was conducted using X-ray diffractometer Empyrean (PANalytical, Almeo, The Netherlands) to determine the crystalline structure of nanoparticles.

Parameters of the measurement:Copper cathode;Emission wavelength 1.54 A;Current 35 mA;Voltage 40 kV;2θ measurement range 10–90°;2θ sampling frequency: 0.01°;

The determination of the gelation point was carried out on a rotary viscometer “Fungilab Expert” (“Fungilab S. A.”, Barselona, Spain), the action of which is based on the use of viscous friction arising in a layer of liquid flowing in the annular gap between the rotating and stationary cylinders.

Parameters of the measurement:Cylinder TL 1–8;Temperature 25 °C;The rotation speed of 0.1–200 rpm [66].

The molecular simulation was carried out in the IQmol molecular editor, and the quantum-chemical calculations of the models were carried out using the QChem software with the following parameters: Calculation—Energy, method—B3LYP, Basis—6–31G*, Convergence—5, Force field—Chemical [67].

To study functional groups in the obtained samples, IR spectroscopy was used. IR spectra were recorded on an FSM-1201 IR-spectrometer with Fourier transform. The measurement range was 400–4400 cm^−1^.

The ζ-potential and active acidity of the medium was determined by acoustic and electroacoustic spectroscopy on a DT-1202 setup (Dispersion Technology, NY, USA) [68]. The method of preparation of solutions with an accurate pH is presented in the previous published work [69].

### 2.3. Treatment of the Hordeum vulgare L. Seeds

In the experiment, seeds of the *Hordeum vulgare* L. culture used were stored for more than 10 years, with reduced morphological and functional characteristics. The seeds of the *Hordeum vulgare* L. culture were collected and identified on the basis of the Stavropol Research Institute of Agriculture (SNIISH, Stavropol, Russian) in 2011. The samples were stored in the biological collection of grain ears, sorghum crops, perennial fodder, and medicinal plants at the Stavropol Research Institute of Agriculture. Seed collection was carried out according to Russian national standard GOST 12036-85 “Seeds of agricultural crops. Acceptance rules and sampling methods”. Appropriate permission has been obtained for the collection of seeds samples.

Previously, the seeds were not subjected to further processing. The seeds were treated with solutions of Mn_x_O_y_ stabilized with L-methionine, L-methionine, and potassium permanganate (KMnO_4_) at various concentrations (0.0005; 0.005; 0.05; 0.5; 5 mg/mL). In the control sample, the seeds were treated with distilled water.

*Hordeum vulgare* L. seeds treated with solutions of various concentrations were placed in a thermostat at a constant temperature of 20 °C without illumination. Each sample contained 100 seeds. A total of 72 h after treatment, the percentage of germinated seeds (germination energy) was calculated, lengths of root and seedling were measured, and the obtained data were compared with the control.

### 2.4. Statistical Data Processing

Statistical processing of the experimental results was carried out using Statistica 12.0 software (StatSoft, Tulsa, OK, USA). To assess the significance of dressing agent’s concentration as a factor affecting the seeds germination percentage, the length of roots, and shoots length, intragroup and intergroup analysis of variance (ANOVA) was carried out. The nature of the distribution of quantitative characteristics was determined using the Shapiro–Wilk test, the equality of variances was determined using the Leuven test. Tukey’s multiple comparison test was used to determine the significant difference between the samples. To compare quantitative characteristics in the intergroup analysis of variance, the Kruskal–Wallis test was used. Quantitative values were assessed using the median (Me) and the 25th and 75th percentiles. The critical level of significance (*p*) was taken equal to 0.05.

## 3. Results and Discussion

### 3.1. Mn_x_O_y_ Nanoparticles Characterization

In the first stage of the research, dried samples of Mn_x_O_y_ nanoparticles stabilized with L-methionine were examined by scanning electron microscopy (SEM) and powder diffractometry. The data obtained are presented in Figure 1 and Figure 2.

The SEM micrographs of Mn_x_O_y_ nanoparticles stabilized with L-methionine (Figure 1) showed the presence of homogeneous aggregates of spherical nanoparticles with an average size of 15–30 nm.

Analysis and interpretation of the diffractogram (Figure 2) showed the presence of low-intensity broadened peaks indicating the existence of an amorphous MnO_2_ phase with a tetragonal crystal lattice having spatial group *I4/m* (unit cell parameters: a—9.8150 Å, b—9.8150 Å, c—2.8470 Å) [70]. The existence of the second phase Mn_3_O_4_, with a tetragonal crystal lattice having spatial group *I41/amd* (unit cell parameters: a—5.7574 Å, b—5.7574 Å, c—9.4239 Å) is also established [71].

To study the mechanism of stabilization of Mn_x_O_y_ nanoparticles stabilized with L-methionine, quantum chemical modeling was carried out in the QChem software [72,73,74]. According to the literature data [75], methionine is oxidized to methionine sulfoxide (Figure 3).

At the first stage of quantum chemical modeling, the models of methionine sulfoxide, MnO_2_ and Mn_3_O_4_ were considered separately. The manganese dioxide and double manganese oxide molecules are represented by sequences of three manganese atoms connected to a certain number of oxygen atoms, which correlates with the valencies of these Mn atoms and, taking into account the coordination field, the free bonds are hydrogenated. The obtained models, electron density distributions, electron density distribution gradients, and molecular orbitals are presented in Figure 4, Figure 5 and Figure 6.

From the data analysis, it was found that the interaction of methionine sulfoxide with manganese oxides can occur through the oxygen of the sulfogroup, the amino group and the carboxyl group. In the next stage of modeling, the variants of the interaction of methionine sulfoxide with manganese oxides are considered. The obtained models and the results of quantum chemical calculations are presented in Table 1 and in the Appendix A.

According to references [76,77,78,79], the highest occupied molecular orbital (HOMO) and the lowest unoccupied molecular orbital (LUMO) are the main orbitals describing the chemical stability of the system. HOMO is directly related to the concept of “ionization potential”. LUMO is related to the concept of “electron affinity”. The energy difference between the LUMO and HOMO orbitals is called the energy gap (ΔE). Based on the data obtained, the value of the chemical hardness (η) was calculated. The chemical hardness characterizes the stability of the system. The chemical hardness is calculated as follows:(1)$$\eta =\frac{1}{2}\times (E_{\mathrm{LUMO}}-E_{\mathrm{HOMO}})$$
where E_LUMO_—LUMO energy; E_HOMO_—HOMO energy.

The analysis of the obtained data showed that the interaction of methionine sulfoxide with manganese oxides is an energetically advantageous process, as evidenced by a decrease in the total energy (E). The interaction of the amino group of methionine sulfoxide with MnO_2_ and Mn_3_O_4_ was found to be the most energetically advantageous (Figure 7 and Figure 8).

To confirm the quantum chemical modeling data, IR spectroscopy of the obtained samples of Mn_x_O_y_ nanoparticles stabilized with L-methionine was performed. The obtained data are presented in Figure 9. 

The analysis of IR spectra of L-methionine and Mn_x_O_y_ nanoparticles stabilized with L-methionine showed that in the range from 2500 cm^−1^ to 3600 cm^−1^ the presence of bands of valence vibrations of bonds is observed as follows: from 2518 cm^−1^ to 2675 cm^−1^—-S-, at 2727 cm^−1^—-CH_2_, from 2882 cm^−1^ to 2858 cm^−1^—-CH_3_, from 2912 cm^−1^ to 2924 cm^−1^—-CH, at 2942 cm^−1^ —-OH, from 3452 cm^−1^ to 3473 cm^−1^—-N-H [80,81].

In the IR–spectrum of L-methionine, characteristic bands of bond deformation vibrations in the range of 400 cm^−1^ to 2200 cm^−1^ are observed as follows: at 880 cm^−1^—deformation vibrations of -C-C- bonds, 930 cm^−1^—vibrations of the ionized thiol group-SH; at 961 cm^−1^—symmetric deformation vibrations of the carboxyl group COO^−^; at 1001 cm^−1^—vibrations of the CH_2_ bond; at 1045 cm^−1^—deformation pendulum vibrations of the ionized amino group NH_3_^+^; at 1082 cm^−1^—symmetric vibrations of the OH^−^ group, the band range from 1101 to 1159 cm^−1^ corresponds to the vibrations of the CH bond; at 1219 cm^−1^—deformation pendulum vibrations of the -CH bond; at 1275 cm^−1^—fan vibrations of the -CH_2_ bond; at 1314 cm^−1^—symmetric vibrations of the group OH^−^; at 1341 cm^−1^—deformation vibrations of the ionized amino group NH_3_^+^; at 1414 cm^−1^—symmetric vibrations of the carboxyl group COO^−^; at 1445 cm^−1^—CH_2_ bond vibrations, in the band range from 1514 cm^−1^ to 1609 cm^−1^ corresponds to symmetric vibrations of the ionized amino group NH_3_^+^; at 1624 cm^−1^—C=C bond vibrations; at 1659 cm^−1^—asymmetric vibrations of the carboxyl group COO^−^; at 1715 cm^−1^—C=O bond vibrations; at 1915 cm^−1^—-C=C bond vibrations, at the range from 1958 cm^−1^ to 2182 cm^−1^ corresponds to the vibrations of the ionized thiol groups -SH [82,83].

In the IR-spectrum of Mn_x_O_y_ nanoparticles, bands characteristic of deformation vibrations are observed in the same range from 400 cm^−1^ to 2200 cm^−1^: the range from 439 cm^−1^ to 779 cm^−1^ corresponds to vibrations of -CH and -CH_2_ bonds, at 880 cm^−1^–vibrations of -C-C bonds, the range from 1281 cm^−1^ to 1341 cm^−1^ corresponds to fan vibrations of the -CH_2_ bond, the range from 1404 cm^−1^ to 1645 cm^−1^ corresponds to symmetric vibrations of the ionized amino group NH_3_^+^, at 1743 cm^−1^–asymmetric vibrations of the carboxyl group COO^−^ [84,85]. It is important to note that in the IR spectrum of the sample there are oscillations in the range from 1058 to 1150 cm^−1^, which are characteristic of scissor oscillations of the S=O group. The presence of this group confirms the oxidation of L-methionine and the formation of methionine sulfoxide [86,87].

It is important to note that in the IR-spectrum of Mn_x_O_y_ nanoparticles, there is a decrease in the intensity of the bands at 1341 cm^−1^ and 1514–1609 cm^−1^, which indicates the formation of a chemical bonds and the interaction of Mn_x_O_y_ nanoparticles with L-methionine through the amino group [88,89].

Then, information was obtained on the effect of the Mn_x_O_y_ sol concentration on the gelation process in this colloidal system.

A graph of the dependence of the dynamic viscosity of the nanoscale Mn_x_O_y_ solutions on its concentration is presented in Figure 10. Analysis of Figure 10 showed that the resulting sample can be characterized as a Newtonian liquid with a gelation point of 0.1 mg/mL. The result is that the dynamic viscosity of solutions of nanoscale Mn_x_O_y_ in the concentration range from 0.0005 to 0.1 mg/mL did not change significantly and was 1–3 MPa∙s. This indicates that at these concentrations, the samples are a free-dispersed colloidal systems (sol). When the concentration of nanoscale Mn_x_O_y_ is 0.1 mg/mL, a sharp jump in the dynamic viscosity is observed, which indicates that the gelation point has been reached and the type of colloidal system has changed. At a concentration of nanosized Mn_x_O_y_ ≥ 0.1 mg/mL, bound-dispersed (gel-cohesive) systems are formed, which can be identified as a Newtonian fluid.

To confirm the obtained data, the hydrodynamic radius of Mn_x_O_y_ nanoparticles solutions was studied before the gelation point (0.05 mg/mL) and after (0.5 mg/mL). The resulting histograms are shown in Figure 11 and Figure 12.

As shown in Figure 11, the distribution of hydrodynamic particle radii in Mn_x_O_y_ samples is monomodal with an average value of 25 nm and a dispersion of 10–100 nm. The analysis of Figure 12 showed that the sample has a bimodal distribution and consists of two fractions. The particles of the first fraction have an average hydrodynamic radius of 25 nm and a dispersion of 10–80 nm, while particles of the second fraction have an average hydrodynamic radius of 512 nm and a dispersion of 100–1000 nm. The data obtained are consistent with the works [90,91]. The results obtained are consistent with the SEM microscopy data.

At the next stage, the effect of the pH of the solution on the ζ-potential of Mn_x_O_y_ nanoparticles stabilized with methionine was investigated. The obtained dependence is shown in Figure 13. It is established that during the transition from an acidic medium to an alkaline one, the nanoparticles “recharge” occurs, i.e., the ζ-potential changes from +48.5 mV to −15.5 mV. It is important to note that an isoelectric point is observed at pH = 8.25. The obtained result is consistent with the work of Neves et al. [92].

As is known, an acidic environment leads to the activation of amino groups of amino acids, and an alkaline environment leads to the activation of carboxyl groups of amino acids [93,94]. Upon protonation of an amino group in an acidic medium, the molecule L-methionine acquires a positive charge (NH_3_^+^) and interacts with the Mn_x_O_y_ micelle, as shown above by the simulation results and IR spectroscopy. With a decrease in hydrogen ions concentration (an increase in pH), the speed of the protonation process slows down, and the equilibrium shifts in the opposite direction, and as a result the charge of the amino groups decreases and becomes zero at the isoelectric point. A further increase in the concentration of OH^−^ ions, above the isoelectric point, leads to a change in the charge of the L-methionine molecule, which becomes negative, due to the deprotonization of carboxyl groups. The scheme of the described chemical transformations is shown in Figure 14.

Thus, an increase in the active acidity leads to a loss of NH_3_^+^ charge and an increase in the desorption of uncharged L-methionine molecules from the surface of Mn_x_O_y_ nanoparticles. As a result, the particles lose their stabilizing shell and begin to coagulate, forming the spatial structure of the gel, which is shown in Figure 15.

### 3.2. Investigation of the Effect of Mn_x_O_y_ Nanoparticles Stabilized with Methionine on Germination of Barley Seeds

The results of the study of the effect of Mn_x_O_y_ nanoparticles stabilized with methionine on the morphofunctional parameters of seeds germination are presented in Table 2 and Table 3 and in the Appendix A.

The analysis of the obtained data showed that 10 years’ storage has reduced morphofunctional characteristics of barley seeds. It was found that the germination energy of control samples was less than 50%. It was also set that the length of seeding in the concentration range of 0.0005–0.5 mg/mL for Mn_x_O_y_ nanoparticles, potassium permanganate KMnO_4,_ and for L-methionine is higher than that of the control sample. Similarly, it is discovered that when the concentration of L-methionine is 5 mg/mL, the length of the *Hordeum vulgare* L. culture seeding takes values lower values than the values of the control sample. There was a statistically significant prevalence of seeding length in samples treated with 0.05 mg/mL Mn_x_O_y_ nanoparticles (32.13 ± 3.40 mm) by 68% compared to the control sample (19.1 ± 2.08 mm). Additionally, Table 3 shows that the effect of Mn_x_O_y_ nanoparticles on the root length of *Hordeum vulgare* L. culture has a nonlinear relation with an extremum at 0.05 mg/mL (34.89 ± 2.34mm). The growth of the root length of samples treated with Mn_x_O_y_ nanoparticles at the point of 0.05 mg/mL is higher by 62.8% (21.43 ± 3.54 mm), 32.7% (26.29 ± 3.22 mm), and 158.9% (13.48 ± 2.66 mm) compared to samples treated with L-methionine, KMnO_4_, and the control sample, respectively.

Analysis of seed germination energy versus substance concentration showed the predominance of the sample treated with Mn_x_O_y_ nanoparticles at the point of 0.05 mg/mL (66.00 ± 2.55%) by 50.0% compared to the control sample (44.00 ± 1.78%) [95], by 10.0% compared to the sample treated with L-methionine (60.00 ± 1.00%), and by 13.8% compared to the sample treated with KMnO_4_ (58.00 ± 1.58%). The data obtained are consistent with the data of other authors. For example, Esper Neto et al. describes the effect of Mn_2_O_3_ nanoparticles on corn seeds. The authors showed that at the concentration of Mn_2_O_3_ nanoparticles of 20 mg/L, the increase in root length was 20% and reached 55 mm [96]. An Iincrease in the root length under the influence of nanoparticles was also observed in the work of Landa et al. [57].

The effect of L-methionine-stabilized Mn_x_O_y_ on the morphological and functional properties of the Hordeum vulgare L. culture is a non-linear dependence (parabolic type) with an extremum at 0.05 mg/mL. (The constructed dependences are presented in the Appendix A. The parabolic type of dependence is due to the fact that at 0.1–5 mg/mL, Mn_x_O_y_ stabilized with L-methionine is a bound-dispersed system (gel), which makes particle diffusion difficult, and the physiological effect on the seeds of the *Hordeum vulgare* L. culture is minimal. At concentrations from 0.0005 to 0.1 mg/mL, the samples constitute a free-dispersed system (sol), which is characterized by high biological activity and accessibility. The described process is schematically shown in Figure 16. Non-aggregated Mn_x_O_y_ nanoparticles stabilized by L-methionine can normalize the level of reception of the essential trace element Mn in the seeds of the culture of *Hordeum vulgare* L., thereby activating metabolic processes. The obtained tendency meets the results of other scientists [58,59].

## 4. Conclusions

In this work, we synthesized of Mn_x_O_y_ nanoparticles stabilized with L-methionine and studied their structure and phase composition of the material were studied. As a result of quantum chemical modeling, the model of stabilization of Mn_x_O_y_ with L-methionine was confirmed by IR spectroscopy. The result of the study of physico-chemical properties of the colloidal system Mn_x_O_y_-L-methionine was proposed as a gelation mechanism.

The effect of Mn_x_O_y_ stabilized with L-methionine on the morphofunctional characteristics of seeds of the *Hordeum vulgare* L. culture is a nonlinear relation with an extremum at 0.05 mg/mL. In our opinion, this effect is related to the correlation of the type of colloidal system of nanoscale Mn_x_O_y_ with the concentration of Mn_x_O_y_ particles. Thus, it was shown that at a concentration of Mn_x_O_y_ < 0.05 mg/mL, the samples are sol-free colloidal systems, and at a concentration of MnxOy > 0.05 mg/mL—gels-cohesive systems.

We found a statistically significant prevalence of seeding length in samples treated with 0.05 mg/mL Mn_x_O_y_ nanoparticles by 68% compared to the control sample. It is shown that the roots growth length of samples treated with Mn_x_O_y_ stabilized with L-methionine at the point of 0.05 mg/mL is higher by 62.8%, 32.7%, and 158.9% compared to samples treated with L-methionine, KMnO_4_, and the control sample, respectively. It was found that the germination energy of seeds treated with Mn_x_O_y_ stabilized with L-methionine at a concentration of 0.05 mg/mL are higher by 50.0% compared to the control sample, by 10.0% compared to the sample treated with L-methionine, and by 13.8% compared to the sample treated with KMnO_4_.

The results of this research allow us to conclude that the use of Mn_x_O_y_ stabilized with L-methionine in the pre-sowing treatment of unconditioned seeds of agricultural crops with low germination rates will activate its metabolism and thereby increase and improve the growth characteristics of young plants, which is essential for agriculture, especially in extreme conditions with adverse weather conditions and climatic change. It is important to note that the proven effectiveness of Mn_x_O_y_ stabilized with L-methionine allows them to be used not only on seeds with low germination rates, but also for seeds with high varietal qualities.

The data obtained on the positive effect of Mn_x_O_y_ stabilized with L-methionine on seeds germination of seeds of the agricultural crop *Hordeum vulgare* L. formed the basis for further experimental studies on other crops.

## Figures and Tables

**Figure 1 nanomaterials-13-01577-f001:**
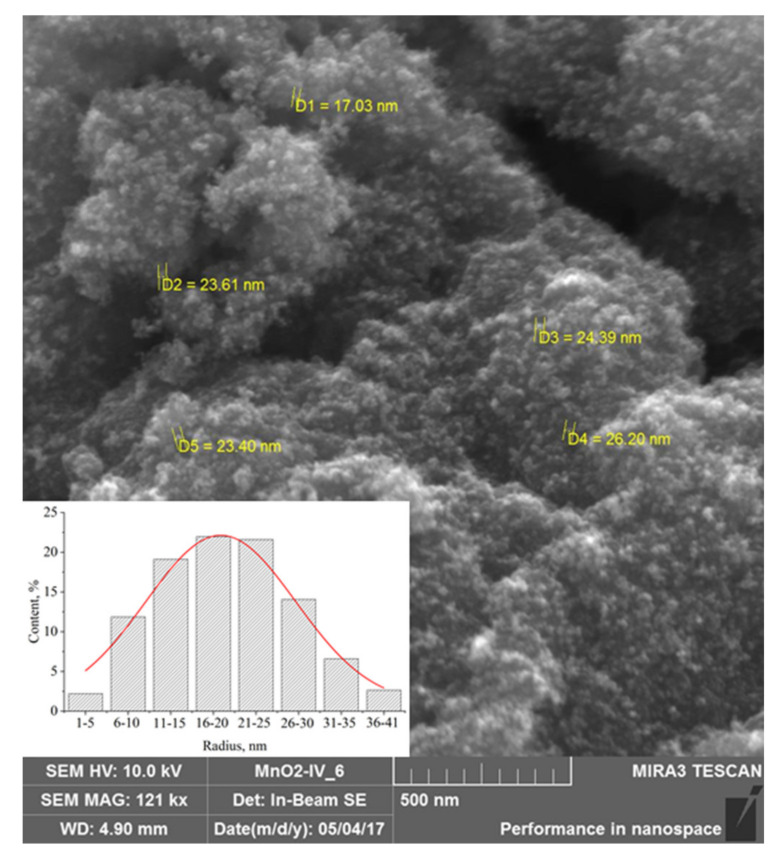
SEM-micrograph of Mn_x_O_y_ nanoparticles stabilized with L-methionine.

**Figure 2 nanomaterials-13-01577-f002:**
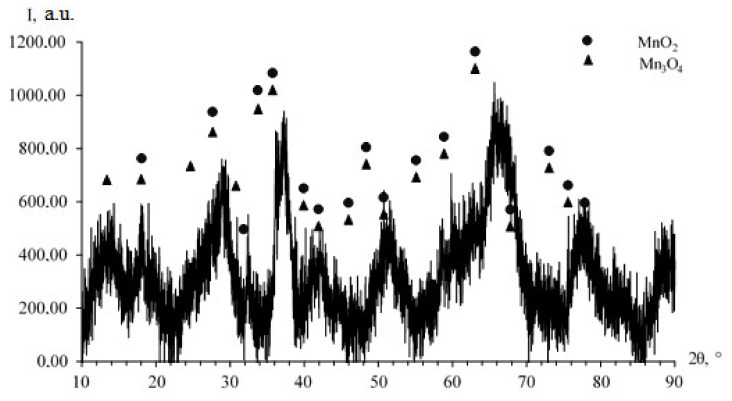
XRD pattern of Mn_x_O_y_ nanoparticles stabilized with L-methionine.

**Figure 3 nanomaterials-13-01577-f003:**
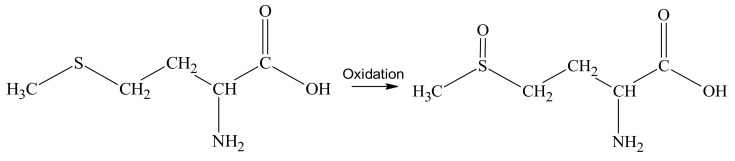
Oxidation of L-methionine.

**Figure 4 nanomaterials-13-01577-f004:**
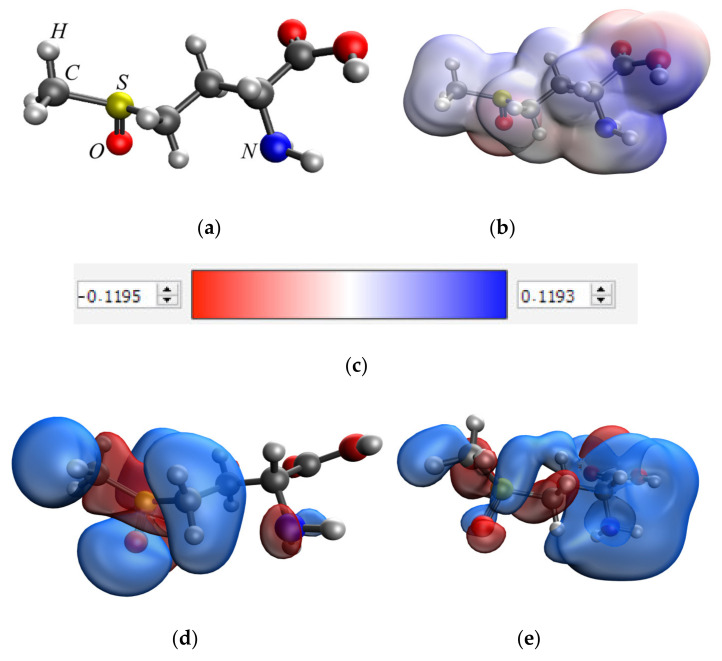
Results of quantum chemical modeling of the methionine sulfoxide molecule: model of the molecule (**a**), electron density distribution (**b**), electron density distribution gradient (**c**), HOMO molecular orbital (**d**), LUMO molecular orbital (**e**).

**Figure 5 nanomaterials-13-01577-f005:**
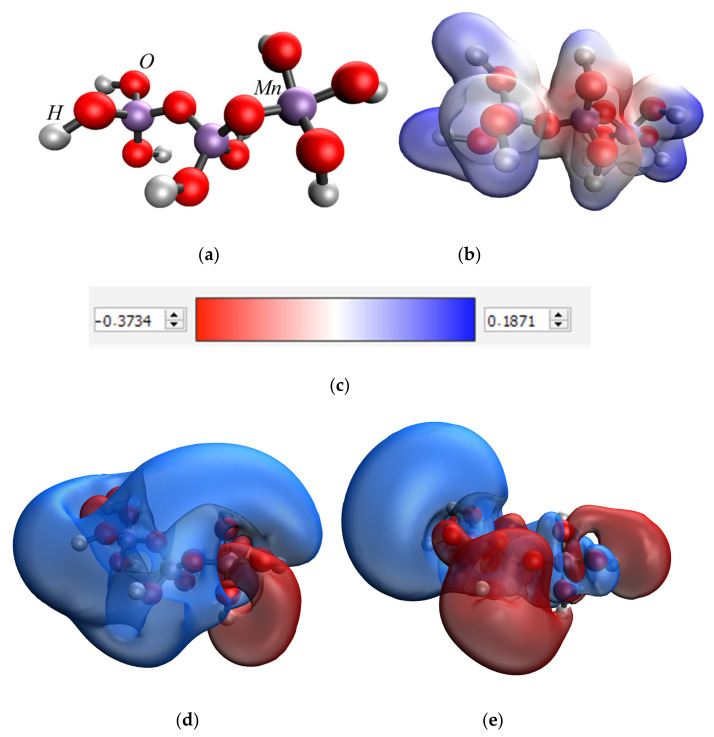
Results of quantum chemical modeling of MnO_2_ molecule: model of the molecule (**a**), electron density distribution (**b**), electron density distribution gradient (**c**), HOMO molecular orbital (**d**), LUMO molecular orbital (**e**).

**Figure 6 nanomaterials-13-01577-f006:**
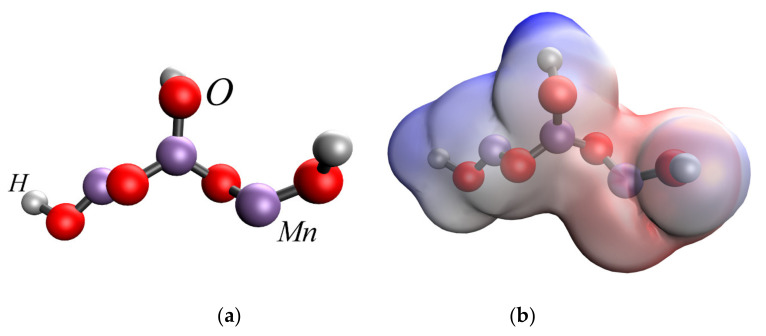
Results of quantum chemical modeling of Mn_3_O_4_ molecule: model of the molecule (**a**), electron density distribution (**b**), electron density distribution gradient (**c**), HOMO molecular orbital (**d**), LUMO molecular orbital (**e**).

**Figure 7 nanomaterials-13-01577-f007:**
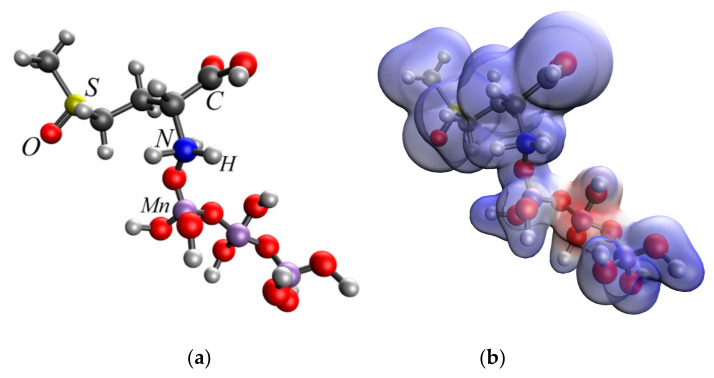
Interaction of the amino group of methionine sulfoxide and MnO_2_: molecular complex model (**a**), electron density distribution (**b**), electron density distribution gradient (**c**), HOMO molecular orbital (**d**), LUMO molecular orbital (**e**).

**Figure 8 nanomaterials-13-01577-f008:**
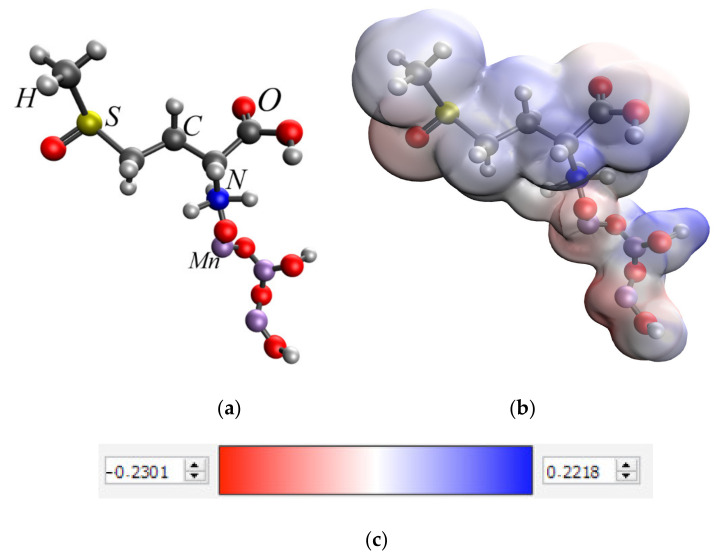
Interaction of the amino group of methionine sulfoxide and Mn_3_O_4_: molecular complex model (**a**), electron density distribution (**b**), electron density distribution gradient (**c**), HOMO molecular orbital (**d**), LUMO molecular orbital (**e**).

**Figure 9 nanomaterials-13-01577-f009:**
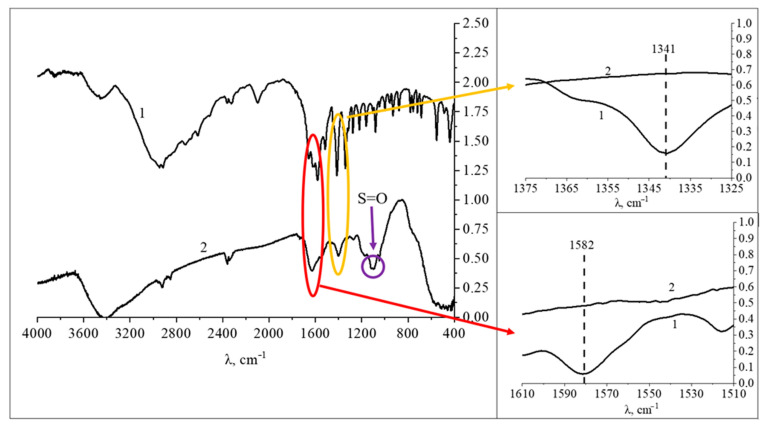
IR-spectra of samples: 1—L-methionine, 2—Mn_x_O_y_ nanoparticles stabilized with L-methionin.

**Figure 10 nanomaterials-13-01577-f010:**
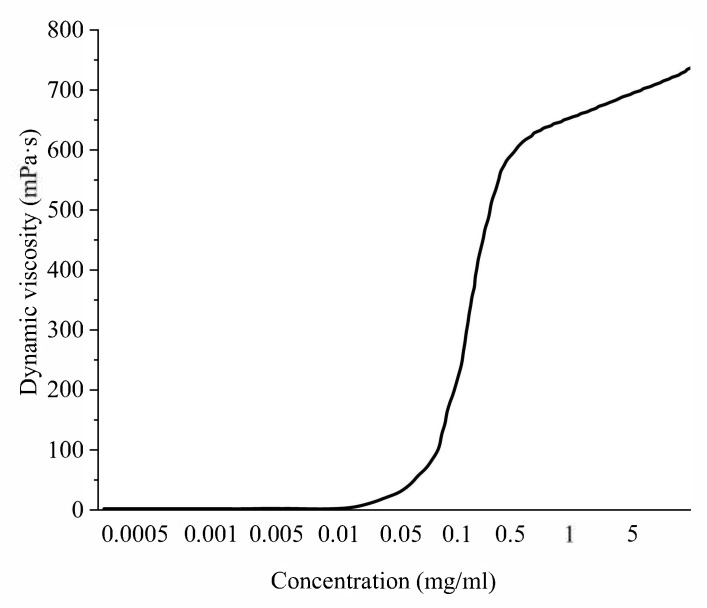
Dependence of the dynamic viscosity of Mn_x_O_y_ colloidal solutions on the concentration of Mn_x_O_y_.

**Figure 11 nanomaterials-13-01577-f011:**
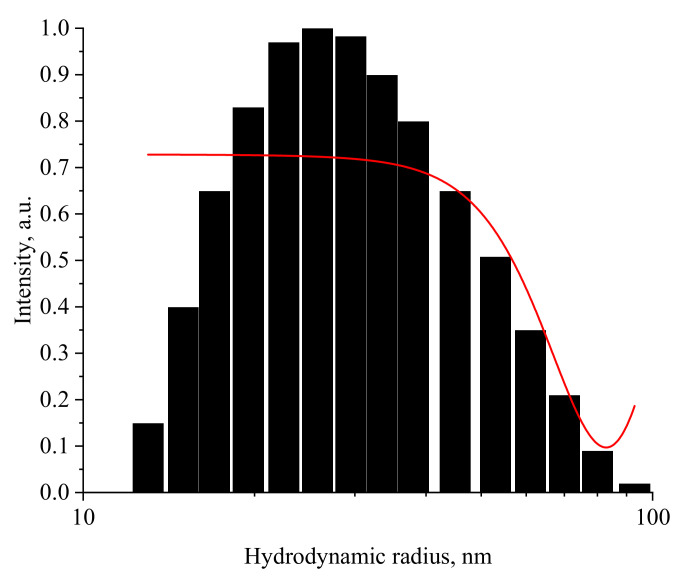
Histogram of the distribution of hydrodynamic radii of Mn_x_O_y_ stabilized with methionine in sol at concentration of 0.05 mg/mL (*X*-axis—average hydrodynamic radius (nm), *Y*-axis—intensity).

**Figure 12 nanomaterials-13-01577-f012:**
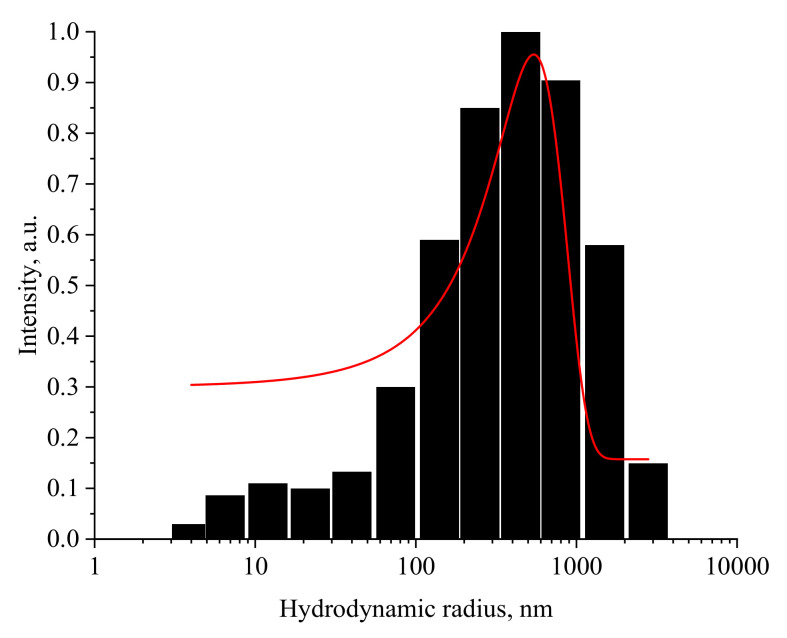
Histogram of the distribution of hydrodynamic radii of Mn_x_O_y_ stabilized with methionine in gel at concentration of 0.5 mg/mL (*X*-axis—average hydrodynamic radius (nm), *Y*-axis—intensity).

**Figure 13 nanomaterials-13-01577-f013:**
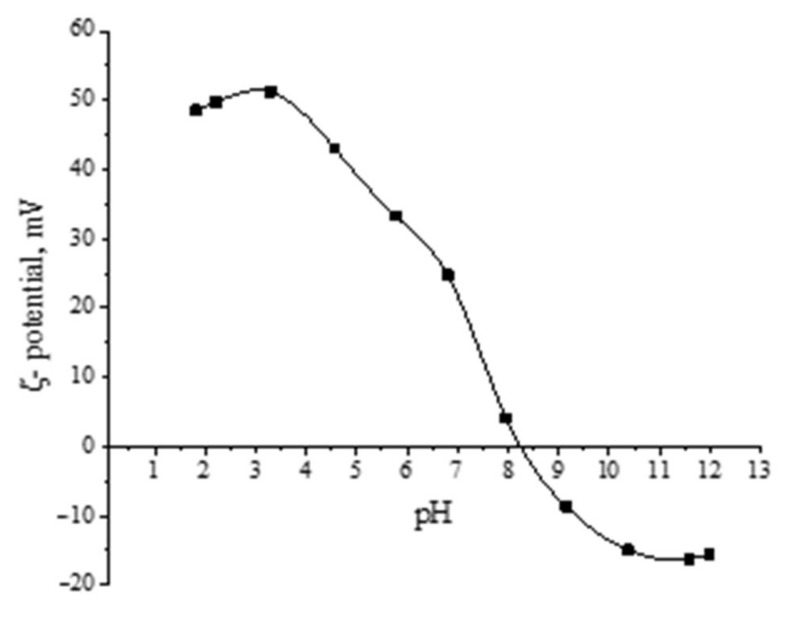
Dependence of the ζ-potential of Mn_x_O_y_ nanoparticles stabilized with methionine on the pH of the solution.

**Figure 14 nanomaterials-13-01577-f014:**
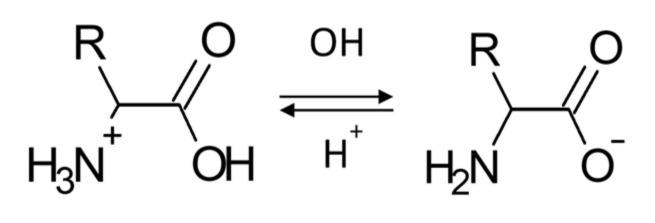
Scheme of the effect of active acidity of the medium on the ionization of L-methionine functional groups.

**Figure 15 nanomaterials-13-01577-f015:**
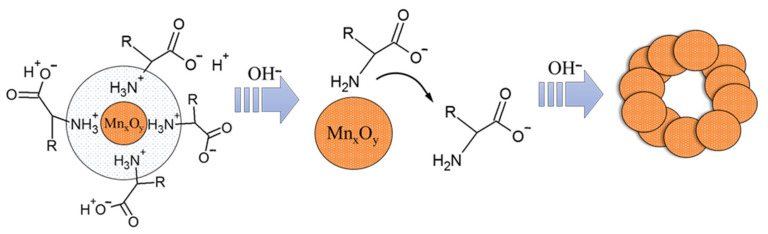
The scheme of Mn_x_O_y_ gel formation.

**Figure 16 nanomaterials-13-01577-f016:**
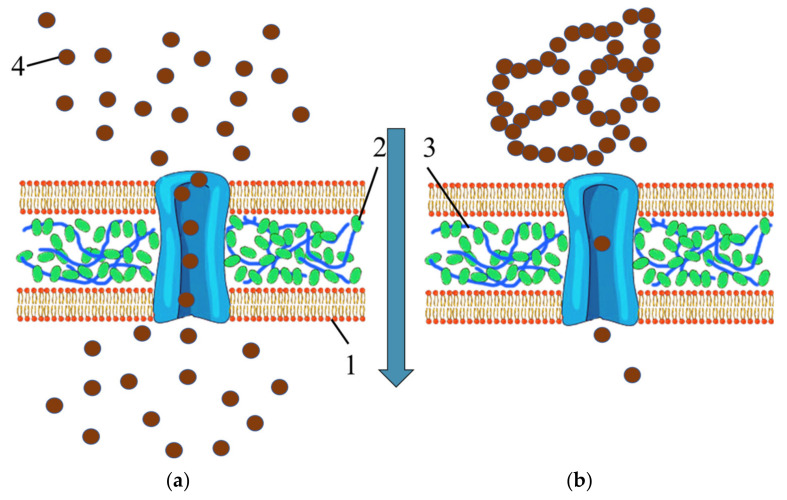
Scheme of diffusion of the Mn_x_O_y_ stabilized with L-methionine in sol-form (**a**) and gel-form (**b**) into the plant cell: 1—Plasma membrane; 2—Cellulose microfibril; 3—Pectin; 4—Mn_x_O_y_ stabilized with L-methionine.

**Table 1 nanomaterials-13-01577-t001:** Results of quantum chemical calculations of the interaction of methionine sulfoxide with MnO_2_ and Mn_3_O_4_.

Index	E, kcal/mol	HOMO	LUMO	η
Methionine Sulfoxide	−875.664	−0.186	−0.008	0.089
MnO_2_	−4200.463	−0.087	−0.046	0.0205
Mn_3_O_4_	−3825.168	−0.115	−0.74	0.205
Interaction of oxygen of the sulfogroup of methionine sulfoxide and the hydroxogroup at the extreme manganese atom of Mn_3_O_4_	−5000.96	−0.086	−0.032	0.027
Interaction of the amino group of methionine sulfoxide and the hydroxo group at the extreme manganese atom in Mn_3_O_4_	−5075.694	−0.109	−0.057	0.026
Interaction of the carboxyl group of methionine sulfoxide and the hydroxo group at the extreme manganese atom of Mn_3_O_4_	−5000.273	−0.103	−0.056	0.0235
Interaction of oxygen of the sulfogroup of methionine sulfoxide and the hydroxogroup of the middle manganese atom in Mn_3_O_4_	−5000.835	−0.06	−0.035	0.0125
Interaction of the amino group of methionine sulfoxide and the hydroxo group of the middle manganese atom in Mn_3_O_4_	−5076.435	−0.088	−0.043	0.225
Interaction of the carboxyl group of methionine sulfoxide and the hydroxo group at the middle atom in Mn_3_O_4_	−5000.29	−0.1	−0.054	0.023
Interaction of oxygen of the sulfogroup of methionine sulfoxide and the hydroxogroup at the extreme manganese atom of MnO_2_	−4624.686	−0.103	−0.051	0.026
Interaction of the amino group of methionine sulfoxide and the hydroxo group at the extreme manganese atom in MnO_2_	−4700.332	−0.116	−0.072	0.022
Interaction of the carboxyl group of methionine sulfoxide and the hydroxo group at the extreme manganese atom of MnO_2_	−4624.352	−0.127	−0.057	0.035
Interaction of oxygen of the sulfogroup of methionine sulfoxide and the hydroxogroup of the middle manganese atom in MnO_2_	−4624.778	−0.064	−0.033	0.0155
Interaction of the amino group of methionine sulfoxide and the hydroxo group of the middle manganese atom in MnO_2_	−4700.374	−0.152	−0.099	0.0265
Interaction of the carboxyl group of methionine sulfoxide and the hydroxo group of the middle manganese atom in MnO_2_	4624.399	−0.136	−0.096	0.02

**Table 2 nanomaterials-13-01577-t002:** Influence of Mn_x_O_y_ stabilized with methionine, KMnO_4,_ and L-methionine concentrations on morphological and functional characteristics of *Hordeum vulgare* L. seeds.

Dressing Agent	Concentration (C), mg/mL	Length of Seedling, mm	Length of Roots, mm	Germination Energy, %
Mn_x_O_y_ stabilized with methionine	Control	19.1 ± 2.08 a	13.48 ± 2.66 c	44.00 ± 1.78 d
	0.0005	23.13 ± 3.89 ab	22.69 ± 3.37 a	51.20 ± 3.56 ab
	0.005	26.25 ± 4.59 b	25.75 ± 3.33 b	55.00 ± 0.71 bc
	0.05	32.13 ± 3.40 c	34.89 ± 2.34 d	66.00 ± 2.55 e
	0.5	25.04 ± 6.16 ab	26.15 ± 4.34 b	57.00 ± 0.70 c
	5	21.96 ± 6.17 a	19.82 ± 2.81 a	50.50 ± 0.84 a
KMnO_4_	Control	19.1 ± 2.08 a	13.48 ± 2.66 a	44.00 ± 1.78 b
	0.0005	21.44 ± 2.61 ab	22.61 ± 3.05 bc	56.00 ± 1.58 a
	0.005	25.00 ± 4.03 c	27.66 ± 3.69 d	58.00 ± 1.00 a
	0.05	26.10 ± 3.99 c	26.29 ± 3.22 cd	58.00 ± 1.58 a
	0.5	22.35 ± 1.67 b	21.47 ± 3.24 b	56.00 ± 0.71 a
	5	19.61 ± 3.00 a	13.78 ± 4.06 a	48.00 ± 1.58 c
L-methionine	Control	19.1± 2.08 a	13.48 ± 2.66 b	44.00 ± 1.78 c
	0.0005	20.56 ± 6.56 ab	24.26 ± 1.80 a	57.2 ± 1.79 ab
	0.005	23.43 ± 4.92 b	22.83± 1.85 a	60.00 ± 1.41 a
	0.05	21.75 ± 8.60 ab	21.43 ± 3.54 a	60.00 ± 1.00 a
	0.5	18.55 ± 4.67 ac	16.38 ± 3.69 b	56.00 ± 1.22 b
	5	14.09 ± 2.60 c	9.70 ± 2.98 c	49.00 ± 0.71 d

The data presented are the mean of three replicates (n = 3) ± standard error. For each dressing agent. Values within the same column marked with different letters are significantly different (*p* < 0.05).

**Table 3 nanomaterials-13-01577-t003:** Comparison of the statistical significance of differences in morphological and functional parameters of seed germination in different groups at concentration of 0.05 mg/mL (Intergroup ANOVA results).

Index	Mn_x_O_y_ (Group 1)			L-Methionine (Group 2)			KMnO_4_ (Group 3)			Control (Group 4)		
	M_e_	25	75	M_e_	25	75	M_e_	25	75	M_e_	25	75
Average length of seeding	31.69 *** ### ●●	29.50	35.00	22.15 §§§	16.00	28.00	25.77 *** §§	23.00	28.00	18.68 §§§ ●●●	18.00	22.00
Average length of roots	34.89 *** ### ●●●	33.75	35.50	21.43 §§§	18.30	25.00	26.29 *** §§§	25.00	29.00	13.48 §§§ ●●●	12.40	15.60
Germination energy	66.00 *** ### ●●●	64.00	68.00	60.00 *** §§§	59.00	61.00	58.00 ** §§§	57.00	59.00	44.00 §§§ ### ●●	42.00	46.00

Significant differences from control (group 3) *p* < 0.01—**, *p* < 0.001—***, differences from group 1 *p* < 0.01—§§, *p* < 0.001—§§§, differences from group 2 *p* < 0.001—###, differences from group 3 *p* < 0.01—●●, *p* < 0.001—●●●. Differences between groups are considered statistically significant at *p* <0.05.

## Data Availability

All data are available upon request from the corresponding author.

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
