# Peer review of "Effect of Mn_x_O_y_ Nanoparticles Stabilized with Methionine on Germination of Barley Seeds (*Hordeum vulgare* L.)"

_nanomaterials, 2023, doi:10.3390/nano13091577_

Round 1

Reviewer 1 Report

Comments

#nanomaterials-2313547: Effect of MnxOy Nanoparticles Stabilized with Methionine on

Germination of Barley Seeds (Hordeum vulgare L.)

This research offers insightful data. The manuscript is well written. Data presentation is very good enough.

Minor revision

1.      If possible Author should provide the image of Hordeum vulgare L seedlings of all treatment to clear show the effect of  MnxOy Nanoparticles

Line 85: Hordeum vulgare L.……….Correct with ………..Hordeum  vulgare 

Author Response

We are grateful to the Reviewer 1 for his/her positive evaluation and for the time devoted to review our manuscript. We have addressed all recommendations as requested. All changes in the manuscript are marked by green. Please see the point-by-point answers below

If possible Author should provide the image of Hordeum vulgare L seedlings of all treatment to clear show the effect of  MnxOy Nanoparticles

Thank you for recommendation! As was suggested by other reviewers, we decided to remove this figure.

Line 85: Hordeum vulgare L.……….Correct with ………..Hordeum  vulgare 

Thank you for suggestion. We decided to use this form of spelling (Hordeum vulgare L) because “L” denotes barley ordinary

Author Response

We are grateful to the Reviewer 2 for his/her positive evaluation and for the time devoted to review our manuscript. All comments were useful and pleased us with the high level of understanding of the topic. We have addressed all recommendations as requested. All changes in the manuscript are marked by green. Please see the point-by-point answers below

On page 5/27: Figure 1 shows the SEM microscopic image of nanoparticles. It would be useful to show a histogram of the size distribution of the nanoparticles and make a Gaussian fit to the histogram data

Thank you very much for recommendation. Histograms are supplemented with Gaussian distributions.

On page 6/27 The authors did not confirm that methionine sulphoxide was identified. showed the oxidation reaction of methionine, but did not show that they had identified it

Thank you for the note. We mentioned that formation of methionine sulfoxide is confirmed by IR spectroscopy data (lines 327 - 330).

On page 13/27 When the concentration of nanoscale MnxOy is 0.1 mg/ml, a sharp jump in dynamic viscosity is observed, indicating that the gelation point has been reached and the type of the colloidal system has changed. The authors did not describe the system studied as a Newtonian or non-Newtonian fluid, nor did they verify which type of system it was. If it is a non-Newtonian fluid, the stress rate must also be given when specifying the dynamic viscosity.

Thank you for your comment. This gel can be identified as a Newtonian liquid. The gelation point is reached at a concentration of 0.1 mg/ml (lines 340-342).

On page 14/27, Figure 11 shows the histogram of the hydrodynamic radius. Again, it would be necessary to make a Gaussian fit to the data and compare it to the sizes of the nanoparticles obtained by the SEM analysis.

Thank you for the comment. Histograms are supplemented with Gaussian distributions

On page 15/27, the same applies to Figure 12 of the nanoparticle histogram

Thank you for the comment. Histograms are supplemented with Gaussian distributions

On page 15/27, the dependence of pH on concentration is shown, and nowhere have the authors indicated where the isoelectric point of the observed nanoparticles is located. This should definitely be rectified

Thank you for the comment. To determine the isoelectric point, an additional experiment was conducted to study the effect of the pH of the solution on the ζ-potential of MnxOy nanoparticles.The resulting dependence is shown in Figure 13. It has been established that during the transition from an acidic medium to an alkaline one, nanoparticles "recharge" occurs, i.e. the ζ-potential changes from +48.5 mV to -15.5 mV. It is important to note that an isoelectric point is observed at pH = 8.4. The obtained result is consistent with the work of Neves et al. (2016).

The section with the study of the effect of concentration has been removed from the text of the article.

On page 17/27 the authors show the dependence of conductivity on concentration, but it would be better to show the dependence of conductivity on pH.

Thank you for recommendation. Considering your and other reviewers’ comments we decided to remove this part from the main text.

Reviewer 3 Report

Comments and suggestions for authors

I would like to make a few comments:

1.      The abstract should be a total of about 200 words maximum. Your abstract has 330 words, so it needs to be shortened.

2.      The first time an abbreviation is used in the text, it must be explained. manganese dioxide (MnxOy) , ......................( KMnO4,).

3.      L 85: I suggest writing the name first in English, and the Latin name in brackets: “Hordeum vulgare L. (barley) seeds“  change to „...barley (Hordeum vulgare L.).

4.      L 1543:     with reduced morphological and functional characteristics“. What characteristics were reduced? You need to supply germination of the seed used ant other morphological and functional characteristics.

5.      Also, in the study, I miss a control where the seeds would be used from a new harvest, not stored for 10 years, and whose properties would not have decreased. Such a control was necessary in this study.

6.       I don't understand why the study was done with old seeds? Please clarify this in the introduction by formulating the relevance of the topic.

7.      L 392: „tested in 3x repetition“ - this information is provided in the Material and Methods section and does not need to be repeated.

8.      If the study is performed in 3 replicates, please provide R05 in Table 2. It is necessary.

9.      What does Figure 17 show? Nothing. What concentration is treated? What does it compare to? Also provide photos of seeds treated with other concentrations of solutions. If not, remove this image.

10.  Table 4. What does it mean „Me“ 25 and 75? Should explain.

11.  Reduce Figures 18-20 and combine them into one picture. The way they are presented now looks very unprofessionally prepared. Also remove the background grid.

12.  The article contains a lot of material. Do you need all the figures? Do they have to be that big?

13.  Conclusions are too long. They must be shortened and presented specifically.

14.  The discussion is very poor, weak, inappropriate. I strongly suggest that you separate the Results from the discussion and simply rewrite the article.

I think the article is not completely prepared. Therefore, I propose to separate the Results from the Discussion and summarize the conclusions. Also pay attention to the Figures.

Author Response

We are grateful to the Reviewer 3 for his/her positive evaluation and for the time devoted to review our manuscript. All comments were useful and pleased us with the high level of understanding of the topic. We have addressed all recommendations as requested. All changes in the manuscript are marked by green. Please see the point-by-point answers below

  1. The abstract should be a total of about 200 words maximum. Your abstract has 330 words, so it needs to be shortened.

Thank you for suggestion. The Abstract was shorted.

  1. The first time an abbreviation is used in the text, it must be explained.manganese dioxide (MnxOy) , ......................( KMnO4,).

Thank you for the comment. It was corrected. We encrypted all the abbreviations at the beginning of the text

  1. L 85: I suggest writing the name first in English, and the Latin name in brackets: “Hordeum vulgareL. (barley) seeds“  change to „...barley (Hordeum vulgare L.).

Thank you for recommendation. We agree with it and corrected it in the text.

  1. L 1543: „with reduced morphological and functional characteristics“. What characteristics were reduced? You need to supply germination of the seed used ant other morphological and functional characteristics.

Thank you for the comment. With prolonged storage of seeds, their morphofunctional properties decrease. That is, for 10-year-old seeds, the germination energy is greatly reduced, which is reflected in the scientific literature, which is described in the introduction

  1. Also, in the study, I miss a control where the seeds would be used from a new harvest, not stored for 10 years, and whose properties would not have decreased.Such a control was necessary in this study.

Thank you for your comment. The team of authors believes that the use of seeds from a new crop is impractical due to the fact that:

  1. I) it is impossible to pick up seeds identical to those that were collected 10 years ago, which is due to possible selective transformations. We cannot state with high accuracy that the seeds that were collected at the time of writing this work will correspond to the seeds that were collected 10 years ago.
  2. II) This correction will entail a change in the course of this work, since it is devoted specifically to the study of "old" seeds that have reduced the energy of germination during storage. If we introduce new seeds, then the work will acquire a new meaning, different from the original one.

We will be able to consider your recommendation in the next works. Thank you for your note.

  1. I don't understand why the study was done with old seeds? Please clarify this in the introduction by formulating the relevance of the topic.

Thank you for the comment. Justification for the use of old seeds is added to the text of the article (lines 64-91)

  1. L 392: „tested in 3x repetition“ - this information is provided in the Material and Methods section and does not need to be repeated.

Thank you for your attentiveness. We removed the duplicated information.

  1. If the study is performed in 3 replicates, please provide R05 in Table 2.It is necessary.

Thank you for recommendation. Due to the fact that the data presented in Table 2 are duplicated in Table 3, table 2 has been removed from the text of the article.

  1. What does Figure 17 show?Nothing. What concentration is treated? What does it compare to? Also provide photos of seeds treated with other concentrations of solutions. If not, remove this image.

Thank you for your attentiveness and nice recommendation We removed fig. 17.

  1. Table 4. What does it mean „Me“ 25 and 75? Should explain.

Thank you for the comment.  25 and 75 are percentiles. A percentile is a value that the value does not exceed with a fixed probability set as a percentage. This is a common practice.

  1. Reduce Figures 18-20 and combine them into one picture.The way they are presented now looks very unprofessionally prepared. Also remove the background grid.

Thank you very much for recommendation. We decided to move these figures to Supplementary.

  1. The article contains a lot of material. Do you need all the figures? Do they have to be that big?

Thank you for the comment. The authors believe that the article is full-fledged, the presented material fully reveals the essence of the study. Figures 18-20 have been moved to the Support section, Table 2 has been deleted, the section with the study of the effect of MnxOy nanoparticle concentration on the pH of the solution has been shortened, Figure 17 was removed.

  1. Conclusions are too long. They must be shortened and presented specifically.

Thank you for recommendation. Conclusions section was shorted.

  1. The discussion is very poor, weak, inappropriate. I strongly suggest that you separate the Results from the discussion and simply rewrite the article.

Thank you for recommendation. The authors believe that this structure of the article fully reveals the essence of the study.

Reviewer 4 Report

The manuscript contains interesting and scientific information regarding the effect of MnxOy nanoparticles stabilized with L-methionine on the morpho-functional characteristics of the Barleys (Hordeum vulgare L.) seeds.

You use “MnxOy nanoparticles” everywhere in the manuscript, which is incorrect. Must be MnxOy stabilized with methionine”. Correct it

In the Abstract and Conclusion must add about the nanoparticle synthesis and measurement.

There is almost no discussion on the effect of MnxOy nanoparticles stabilized with methionine on the morpho-functional characteristics of the Hordeum vulgare L. seeds.

Table 2 is superfluous because the average values of the control variant are given in Table 3.

Why you do not have Photos of the Hordeum vulgare L. seed treated with KMnO4 and methionine and control?

The data presented in Figures 18, 19 and 20 are the same as the data in Table 3. Choose to present only one - figures or table.

All remarks are made in the text.

The manuscript could be accepted for publication after major revision.

Author Response

We are grateful to the Reviewer 4 for his/her positive evaluation and for the time devoted to review our manuscript. All comments were useful and pleased us with the high level of understanding of the topic. We have addressed all recommendations as requested. All changes in the manuscript are marked by green. Please see the point-by-point answers below

You use “MnxOnanoparticles” everywhere in the manuscript, which is incorrect. Must be MnxOy stabilized with methionine”. Correct it

Thank you for recommendation. It was accepted and corrected.

In the Abstract and Conclusion must add about the nanoparticle synthesis and measurement.

Thank you for recommendation. It was accepted and corrected.

There is almost no discussion on the effect of MnxOy nanoparticles stabilized with methionine on the morpho-functional characteristics of the Hordeum vulgare L. seeds.

Thank you for recommendation. It was accepted and corrected.

Table 2 is superfluous because the average values of the control variant are given in Table 3.

Thank you for the comment. During revision, table 2 was removed from the text.

Why you do not have Photos of the Hordeum vulgare L. seed treated with KMnO4 and methionine and control?

Thank you for the comment. During revision, figure 17 was removed from the text

The data presented in Figures 18, 19 and 20 are the same as the data in Table 3. Choose to present only one - figures or table.

Thank you for recommendation. Figures 18-20 were moved to Supplementary.

Reviewer 5 Report

The impact of nanomaterials on plant seed germination is currently a research hotspot. The research topic of this paper is correct, and many valuable results and conclusions have been obtained. Suggest publishing this article after minor modification.

1, The scale in Figure 1 is too large, why is the magnification of the electron microscope no longer larger?

2, There are too many Figures in this article, and many of them can be included in the support information.

3, Figure 17 has no meaning in the text.

4, The latest literature on the impact of nanomaterials on plant seed germination needs to be discussed and cited extensively.

AgNPs-Triggered Seed Metabolic and Transcriptional Reprogramming Enhanced Rice Salt Tolerance and Blast Resistance. ACS NANO, 2022

Effect of Silica-Based Nanomaterials on Seed Germination and Seedling Growth of Rice (Oryza sativa L.). NANOMATERIALS, 2022

Microplastic and Nanoplastic Interactions with Plant Species: Trends, Meta-Analysis, and Perspectives. ENVIRONMENTAL SCIENCE & TECHNOLOGY LETTERS, 2022

Author Response

We are grateful to the Reviewer 5 for his/her positive evaluation and for the time devoted to review our manuscript. All comments were useful and pleased us with the high level of understanding of the topic. We have addressed all recommendations as requested. All changes in the manuscript are marked by green. Please see the point-by-point answers below

1, The scale in Figure 1 is too large, why is the magnification of the electron microscope no longer larger?

Thank you for the comment. Unfortunately, the microscope available to the authors team does not allow obtaining high-resolution SEM images

2, There are too many Figures in this article, and many of them can be included in the support information.

Thank you for recommendation. We decided to remove figure 17. Figures 18-20 were moved to Supplementary.

3, Figure 17 has no meaning in the text.

It was removed.

4, The latest literature on the impact of nanomaterials on plant seed germination needs to be discussed and cited extensively.

AgNPs-Triggered Seed Metabolic and Transcriptional Reprogramming Enhanced Rice Salt Tolerance and Blast Resistance. ACS NANO, 2022

Effect of Silica-Based Nanomaterials on Seed Germination and Seedling Growth of Rice (Oryza sativa L.). NANOMATERIALS, 2022

Microplastic and Nanoplastic Interactions with Plant Species: Trends, Meta-Analysis, and Perspectives. ENVIRONMENTAL SCIENCE & TECHNOLOGY LETTERS, 2022

Thank you for suggestion. All sources were useful, we referred them in the text.

Round 2

Reviewer 4 Report

All remarks are made in the text.

The manuscript could be accepted for publication after minor revision.

Author Response

The authors express their graditude to Reviewer 4 for his/her time and constructive suggestions and recommendations that helped to improve the quality of the manuscript. All new corrections in the revised manuscript are marked by yellow.